# CCL4 Regulates Eosinophil Activation in Eosinophilic Airway Inflammation

**DOI:** 10.3390/ijms232416149

**Published:** 2022-12-18

**Authors:** Hanh Hong Chu, Yoshiki Kobayashi, Dan Van Bui, Yasutaka Yun, Linh Manh Nguyen, Akitoshi Mitani, Kensuke Suzuki, Mikiya Asako, Akira Kanda, Hiroshi Iwai

**Affiliations:** 1Department of Otorhinolaryngology, Head and Neck Surgery, Kansai Medical University, Osaka 573-1010, Japan; 2Allergy Center, Kansai Medical University Hospital, Osaka 573-1010, Japan; 3Department of Allergy and Clinical Immunology, Hanoi Medical University, Hanoi 116177, Vietnam

**Keywords:** airway inflammation, CCL4, eosinophilic chronic rhinosinusitis, eosinophils

## Abstract

Eosinophilic chronic rhinosinusitis (ECRS) is a refractory airway disease accompanied by eosinophilic inflammation, the mechanisms of which are unknown. We recently found that CCL4/MIP-1β—a specific ligand for CCR5 receptors—was implicated in eosinophil recruitment into the inflammatory site and was substantially released from activated eosinophils. Moreover, it was found in nasal polyps from patients with ECRS, primarily in epithelial cells. In the present study, the role of epithelial cell-derived CCL4 in eosinophil activation was investigated. First, CCL4 expression in nasal polyps from patients with ECRS as well as its role of CCL4 in eosinophilic airway inflammation were investigated in an in vivo model. Furthermore, the role of CCL4 in CD69 expression—a marker of activated eosinophils—as well as the signaling pathways involved in CCL4-mediated eosinophil activation were investigated. Notably, CCL4 expression, but not CCL5, CCL11, or CCL26, was found to be significantly increased in nasal polyps from patients with ECRS associated with eosinophil infiltration as well as in BEAS-2B cells co-incubated with eosinophils. In an OVA-induced allergic mouse model, CCL4 increased eosinophil accumulation in the nasal mucosa and the bronchoalveolar lavage (BALF). Moreover, we found that CD69 expression was upregulated in CCL4-stimulated eosinophils; similarly, phosphorylation of several kinases, including platelet-derived growth factor receptor (PDGFR)β, SRC kinase family (Lck, Src, and Yes), and extracellular signal-regulated kinase (ERK), was upregulated. Further, CCR5, PDGFRβ, and/or Src kinase inhibition partially restored CCL4-induced CD69 upregulation. Thus, CCL4, which is derived from airway epithelial cells, plays a role in the accumulation and activation of eosinophils at inflammatory sites. These findings may provide a novel therapeutic target for eosinophilic airway inflammation, such as ECRS.

## 1. Introduction

Eosinophilic chronic rhinosinusitis (ECRS) is a type of chronic rhinosinusitis with nasal polyps in which inflammatory cells, particularly eosinophils, are infiltrated in the paranasal sinus mucosa and/or nasal polyps [1,2,3]. In Japan, ECRS is diagnosed using the Japanese Epidemiological Survey of Refractory Eosinophilic Chronic Rhinosinusitis (JESREC) scoring system and mucosal eosinophil counts [4,5]. Notably, the accumulation of infiltrated eosinophils into the nasal mucosa is associated with a reduced local response to corticosteroids and has been identified as a prognostic factor of polyp recurrence following endoscopic sinus surgery (ESS) [4,5,6]. Therefore, eosinophils that are locally infiltrated into the inflammatory site may correlate with disease state and be a therapeutic target for ECRS.

Nasal polyps are formed in ECRS because of the overproduction of inflammatory mediators due to mucosal epithelial damage associated with eosinophil activation as well as prolonged inflammation-mediated remodeling [7,8]. Inflamed epithelial cells release various proinflammation type 2 cytokines and chemokines, including the CC chemokine ligand CCL11 (Eotaxin-1), CCL13 (MCP-4), CCL24 (Eotaxin-2), and CCL26 (Eotaxin-3), which further stimulate the migration of leukocyte subsets, including eosinophils, to specific sites [9,10,11]. Furthermore, matured eosinophils migrate into tissues and express functional receptors for chemoattractants, cytokines, and growth factors [12,13]. Notably, eosinophils are classified into two subpopulations: resident and inducible eosinophils. In a steady-state condition, resident eosinophils reside in the mucosal tissue and contribute to metabolic and immune homeostasis [14]. However, inducible eosinophils are recruited in the inflamed tissue by chemoattractants, such as CCL5 (RANTES), CCL11, CCL24, and CCL26, and they express activation markers, such as CD69, and release inflammatory mediators, including cytokines, chemokines, specific granules (major basic protein [MBP], eosinophil-derived neurotoxin, eosinophil cationic protein, and eosinophil peroxidase) [15,16]. In addition, the co-culture of eosinophils with human bronchial epithelial cells, such as BEAS-2B, increase the release of proinflammatory cytokines and chemokines [17,18,19,20]. Therefore, the interaction between eosinophils and airway epithelial cells may play a role in ECRS pathogenesis.

CCL4 (MIP-1β) is a CC chemokine implicated in immune cells trafficking, specifically to CCR5 receptors [21,22,23,24]. Recently, CCL4, CCL5, CCL11, and CCL26 expressions were found to be higher in nasal polyps than in uncinate process tissue in the same patients with ECRS [6,24]. Notably, CCL4 was implicated in the recruitment of eosinophils into the inflammatory site, and of the eosinophilic chemokines, only CCL4 was substantially released from activated eosinophils [24]. However, the functional role of CCL4 in ECRS pathogenesis remains unknown.

In the present study, we aimed to determine the role of CCL4 released from airway epithelial cells in ECRS as well as its functional role in eosinophil accumulation and activation.

## 2. Results

### 2.1. CCL4 Expression Increases in Airway Epithelial Cells under Eosinophilic Inflammation

To better understand the mechanism of eosinophil recruitment at inflammatory sites, nasal polyps were first examined as the main source of CCL4. Immunofluorescence staining revealed that CCL4 was highly expressed in the area of epithelial cell adhesion molecule (EpCAM)-positive nasal epithelial cells. Furthermore, CCL4 levels were significantly higher in nasal polyps with rich eosinophils (>100/high power field [HPF]) than in those with poor eosinophils (<100/HPF) or without eosinophils, and they were positively correlated with eosinophil count in nasal polyps and with peripheral eosinophil number (Figure 1A). The Characteristics of patients are presented in Appendix A. Notably, mRNA expression of *CCL4*, but not other well-known eosinophil-associated chemokines, such as *CCL5*, *CCL11*, and *CCL26*, was significantly higher in nasal polyps of patients with ECRS than in uncinate tissues of the same patients (Figure 1B). To further elucidate the interaction between eosinophils and airway epithelial cells, we investigated the expression of eosinophilic chemokines in BEAS-2B cells coincubated with purified eosinophils from human peripheral blood. The *CCL4* mRNA expression, but not *CCL5*, *CCL11*, or *CCL26* expression, was significanlty higher in BEAS-2B cells coincubated with eosinophils than in those not coincubated with eosinophils (Figure 1C,D). Altogether, the interaction between eosinophils and airway epithelial cells may enhance CCL4 production from epithelial cells.

### 2.2. CCL4 Enhances Eosinophil Accumulation into the Local Allergic Inflammatory Site

To confirm CCL4’s role in eosinophilic airway inflammation, murine recombinant CCL4 was intranasally administered in a murine model of ovalbumin (OVA)-induced airway inflammation. Further, after OVA challenges and CCL4 administration, eosinophil accumulation in bronchoalveolar lavage fluid (BALF; Figure 2A) and nasal tissues (Figure 2B,C) increase significantly. Therefore, CCL4 may play a role in eosinophilic accumulation.

### 2.3. CCL4 Activates Eosinophils in the Airway

In addition to CCL4, the level of soluble CD69—a marker of eosinophil activation—was elevated in the mucin and serum of patients with refractory ECRS compared with the serum from healthy volunteers. Notably, these levels were significantly higher in mucin than in serum (Appendix A). Furthermore, CCL4 levels were positively correlated with soluble CD69 levels (Appendix A), suggesting that CCL4 is associated with eosinophil activation. Importantly, CCL4 upregulated CD69 expression in eosinophils, and co-incubation with BEAS-2B cells enhanced this upregulation (Figure 3A). In contrast, CCL4 had no effect on eosinophilic viability (Appendix A). Moreover, the expressions levels of CCL4 and CCR5 (a specific receptor of CCL4) increased in eosinophils coincubated with BEAS-2B cells, indicating that eosinophilic activation has a synergistic effect when co-incubated with BEAS-2B cells (Figure 3B,C). Maraviroc—a CCR5 antagonist—partially inhibited CCL4-induced CD69 expression in eosinophils (Figure 3D). Therefore, other signaling pathways may be involved in the CCL4-mediated activation of eosinophils.

### 2.4. Platelet-Derived Growth Factor Receptor (PDGFR)β and Src Family Are Involve in CCL4-Mediated Eosinophil Activation

Other signaling pathways involved in CCL4-mediated CD69 upregulation in eosinophils were investigated. CCL4 stimulation enhanced PDGFRβ, Src kinase family (Lck, Src, and Yes), and extracellular signal-regulated kinase 1/2 (ERK1/2) phosphorylation (Figure 4A). Interestingly, immunofluorescence staining revealed that CCR5, PDGFRβ, and Src were colocalized in eosinophils (Figure 4B). Furthermore, imatinib (a PDGFRβ inhibitor) and dasatinib (a Src inhibitor) significantly reduced CCL4-mediated CD69 upregulation. In addition, imatinib inhibited CD69 expression more than dasatinib, indicating that PDGFRβ is mainly involved in CCL4-mediated eosinophil activation (Figure 4C). Altogether, PDGFRβ may be transactivated via Src phosphorylation after CCR5 activation by CCL4, resulting in eosinophil activation and migration (Figure 5).

## 3. Discussion

The present study demonstrated that CCL4 was released by epithelial cells, and it was significantly increased in eosinophil-rich nasal polyps from patients with ECRS; moreover, epithelial cell-induced CCL4 was correlated to the number of infiltrated eosinophils. In addition, serum CCL4 levels were higher in patients with ECRS than in healthy controls, and mucinous CCL4 levels were remarkably elevated in patients with refractory ECRS. These findings are consistent with findings of previous studies on the relationship between eosinophil counts and CCL4 level in patients with eosinophilic pneumonia and otitis media [24,25,26]. Although activated eosinophils secret CCL4, they are not a major source of CCL4 [24,27,28]. Furthermore, co-incubating eosinophils with BEAS-2B cells enhanced CCL4 secretion; thus, CCL4 may be involved in eosinophilic airway inflammation and may be a clinical marker for ECRS and other eosinophilic inflammatory diseases.

Eosinophils infiltrate into airway inflammatory tissue via cytokines and chemokines from the bloodstream and are involved in ECRS pathogenesis via activated eosinophil-derived proinflammatory mediators [13,29,30]. In an OVA-induced allergy murine model, CCL4 increased eosinophil accumulation in the nasal mucosa and BALF. Interestingly, the infiltration of inducible eosinophil was increased in the inflamed model only after CCL4 administration given that multiple activated eosinophils accumulate at the local inflammatory site [31]. The expression of CCL4 and other eosinophilic chemoattractants (CCL5, CCL11, and CCL26) was reported to be increased in nasal polyps compared with that in uncinate process tissues of the same patients with ECRS [6,32,33,34]. However, compared with the other chemoattractants in this study, CCL4 was significantly increased in eosinophil-rich nasal polyps and was expressed in bronchial epithelial cells coincubated with eosinophils. Notably, CCL4 induces eosinophil activation by upregulating CD69—a marker of eosinophil activation in inflammatory sites [13,29,30]. Furthermore, soluble CD69 levels were significantly higher in mucin than in serum from patients with ECRS. Thus, CD69 was positively correlated to CCL4, indicating that CCL4-mediated eosinophil activation is associated with eosinophilic airway inflammation.

CCR5, which is expressed in leukocyte subsets, such as T cells, macrophages, and eosinophils, play an important role in airway inflammation [35,36,37,38]. CCL4 is a specific CCR5 ligand [24,39], but a selective CCR5 antagonist only partially inhibited the CCL4-mediated enhancement of CD69 in eosinophils. In our previous study, CCR5 antagonist almost completely inhibited eosinophil chemotaxis to CCL4 [24]. CCL4 activates the cellular signaling pathway via transmembrane proteins coupled to G proteins (GPCRs), thereby transactivating other transmembrane receptors, such as the epidermal growth factor receptor, pattern recognition receptors, and receptor tyrosine kinase (RTKs) [40,41,42]. Indeed, CCL4 induced the phosphorylation of several kinases in eosinophils, including PDGFRβ, the Src kinase family (Lck, Src, and Yes), and ERK signaling. Furthermore, compared with Src inhibition, PDGFRβ inhibition significantly restored CCL4-mediated eosinophil activation to a greater extent, whereas Src inhibition only partially restored it. PDGFRβ, an RTK member, regulates biological activities, such as cell migration, including that of eosinophils [43,44,45]. PDGFRβ inhibitors reduce the eosinophil count in eosinophil-associated disorders. In mice with OVA-induced asthma, treatment with imatinib significantly reduced the number of eosinophils around the airways [46,47]. Src family kinases—a group of membrane-associated non-RTKs—play an important role in signaling transduction pathways that promote cell proliferation and migration [48]. Furthermore, GPCR stimulation activates RTKs via tyrosine phosphorylation, which is involved in eosinophil activation [49,50]. RTK transactivation is another important pathway that links between GPCRs and MAPKs [42], indicating that CCR5 activates the Src/PDGFRβ signaling pathway and exerts a biological activity in eosinophils via signal transduction, such as kinase (e.g., ERK) phosphorylation (Figure 5).

## 4. Materials and Methods

### 4.1. Cell Preparation

Eosinophils were isolated with >98% purity from human peripheral blood samples of healthy volunteers with mild eosinophilia (approximately 4–8% of total white blood cells) using negative selection via a MACS system and an eosinophil isolation kit (Miltenyi Biotec, Bergish Gladbach, Germany). After erythrocyte lysis, leukocytes were collected from the human peripheral blood sample and used in some experiments. Furthermore, the human bronchial epithelial cell line BEAS-2B was obtained from the European Collection of Authenticated Cell Culture (Salisbury, UK) and coincubated with purified eosinophils overnight (for 20–24 h), as appropriate. Subsequently, the eosinophils were stimulated with recombinant human CCL4 (Abcam, Cambridge, UK) and pretreated with CCR5 antagonist (Maraviroc; Cayman chemical, Ann Arbor, MI, USA), PDGFRβ inhibitor (Imatinib; Selleck, Houston, TX, USA), or Src family inhibitor (Dasatinib; Selleck), as appropriate. This study was approved by the local ethics committee of Kansai Medical University (approval number: KanIRin1313). All participants in this study provided written informed consent.

### 4.2. Quantitative RT-PCR

RNA was extracted from nasal polyp or uncinated tissues collected from patients with ECRS or from BEAS-2B cells using the RNeasy Mini Kit (Qiagen, Hilden, Germany). The cDNA was then reverse-transcribed using Perfect Real Time (Takara, Shiga, Japan), and qPCR was performed using a QuantiTect SYBR Green PCR kit (Qiagen) on a Rotor-Gene Q (Qiagen). Appendix A describes the amplification primers (5′-3′). Subsequently, relative gene expression was analyzed using the 2^−ΔΔCt^ method, with a GAPDH as a reference.

### 4.3. Immunofluorescence Staining

Nasal polyps were formalin-fixed and paraffin-embedded following endoscopic sinus surgery under general anesthesia. After deparaffinization, rehydration, and proteinase K-induced antigen retrieval, sections were blocked and stained with an epithelial cell marker anti-EpCAM (Cell signaling Technology, Danvers, MA, USA), an eosinophil marker MBP (Bio-Rad, Hercules, CA, USA), and CCL4 (Bioss, Woburn, MA, USA), followed by Alexa 488 goat anti-rabbit antibody and Alexa 647 goat anti-mouse antibody (Jackson, PA, USA). The intensity ratio of CCL4 to EpCAM was calculated using ImageJ 11.24.13. Briefly, all CCL4 or EpCAM signals were labeled, and the intensity was evaluated using ImageJ (a single signal was reversed to 8-bit). BEAS-2B cells coincubated with purified eosinophils or eosinophils were fixed with 4% paraformaldehyde, permeabilized, and, subsequently, blocked. The cells were then incubated with the primary antibodies, and CCL4 (Bioss), CCR5 (Abcam), PDGFRβ (Cell signaling Technology, Woburn, MA, USA), and Src (Bioss) were evaluated, as previously described. Additionally, control antibodies and Hoechst staining (Invitrogen, Paisley, UK) were included in each experiment, and slides were visualized using FV3000 confocal microscopes (Olympus, Tokyo, Japan).

### 4.4. Flow Cytometric Analysis

The purified eosinophils or leukocytes were stained with antibodies conjugated with CD125 (clone: A14), CD16b (clone: B73.1), Siglec8 (clone: 837535) (BD Biosciences, New Jersey, USA), CD3 (clone: OKT3), CD19 (clone: SJ25C1), Siglec8 (clone: 7C9), and CD69 (clone: FN50) (BioLegend, San Diego, CA, USA). Canto II or Aria III (BD Biosciences) and Flowjo software (BD Biosciences) were used to acquire and analyze data, respectively.

### 4.5. Cell Survival

The cell viability was assessed using double staining with 7-amino-actinomycin and Annexin (BD Pharmingen, Franklin Lakes, NJ, USA).

### 4.6. Western Blot

The cell protein extracts were prepared eusing M-PER lysing buffer (Thermo Fisher Scientific, Rockford, IL, USA) with freshly added complete protease inhibitors (Cell Signaling Technology). The protein concentrations were then determined using the BCA Protein Assay Kit (Thermo Fisher Scientific, Rockford, IL, USA). Thereafter, the protein extract was separated using SDS-PAGE (Bio-Rad, Hercules, CA, USA) and detected using western blot analysis using the Odyssey infrared imaging system (LI-COR Bioscience, Lincoln, NE, USA) according to the manufacturer’s instruction. A rabbit polyclonal antibody to CCR5 (GeneTex, Irvine, CA, USA) was used as a primary antibody, and β-actin was used as a loading control.

### 4.7. Human Phosphor-Kinase Array

The Human Phospho-Kinase Array Kit was purchased from R&D Systems (Minneapolis, MN, USA), and the protocol was followed according to the recommendations in the vendor’s handbook.

### 4.8. Animals

Female Balb/c mice (age, 6–8 weeks) purchased from Shimizu Experimental Material (Kyoto, Japan) were housed in a specific pathogen-free animal facility with a regular 12-h light/dark cycle at an approriate temperature and humidity. All experimental procedures were approved by the Animal Care and Use Committee of Kansai Medical University (18-082).

### 4.9. Sensitization and Airway Challenge

On Days 0 and 14, the mice were immunized with an intraperitoneal injection of 50 μg OVA (OVA Grade V; Sigma-Aldrich, St. Louis, MO, USA) emulsified in 1mg of alum (Aluminium Hydroxide Gel Adjuvant; InvivoGen, San Diego, CA, USA) for a total volume of 200 μL, followed by an OVA challenge (1% OVA in phosphate buffered saline [PBS] for 30 min) on Day 25 and 26. These mice were then intranasally administered recombinant murine CCL4 (PeproTech, Cranbury, NJ, USA) or PBS as a control on Days 21–25. After 24 h of the last challenge, the tissues and cells were extracted, and BALF was collected through the tracheal tube with 1 mL of PBS. Trimmed heads, including the nasal cavity, were prepared for paraffin sectioning, and histology was examined using HE staining.

### 4.10. CCL4 and Soluble CD69 Immunoassay

The CCL4 and soluble CD69 levels in serum and mucin supernatants were measured using ELISA kits for MIP-1β/CCL4 (R&D Systems, Minneapolis, MN, USA) and soluble CD69 (Aviscera Bioscience, Santa Clara, CA, USA). Serum samples were collected from healthy volunteers and patients with refractory ECRS. Furthermore, mucin samples from patients with refractory ECRS were dissolved in PBS containing 0.1% dithiothreitol.

### 4.11. Statistical Analysis

The Mann–Whitney *U* test or paired *t*-test was used to compare the data of the two groups. The correlation coefficients were calculated using the Spearman’s rank method. Other data were analyzed by one-way analysis of variance with an adjusted post hoc test for multiple comparisons, as appropriate. At *p*-values of less than 0.05, differences were considered statistically significant. Descriptive statistics values were expressed as the mean ± standard error of the mean (SEM).

## 5. Conclusions

CCL4 might be associated with eosinophil accumulation in the airway inflammatory site, and it may stimulate the Src/PDGFRβ signaling pathway via CCR5, resulting in CD69 upregulation. Further investigation of CCL4-mediated eosinophil function is required to clarify CCL4’s role and explore its potential as a therapeutic target in eosinophilic airway inflammation.

## Figures and Tables

**Figure 1 ijms-23-16149-f001:**
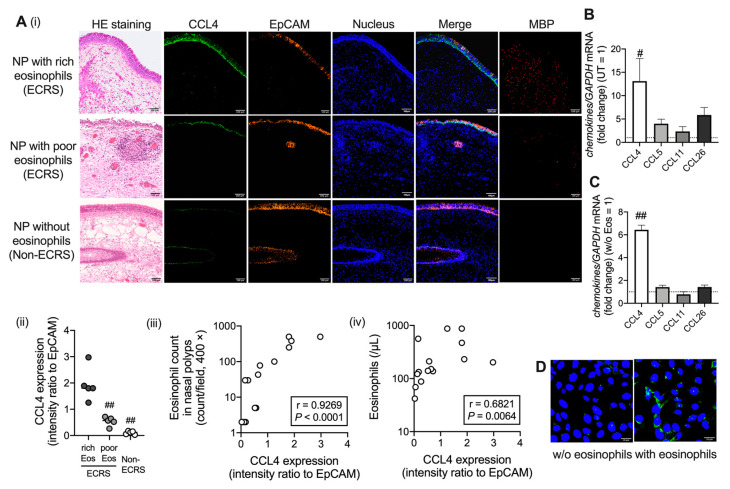
CCL4 expression in airway epithelial cells. (**A**) Immunofluorescence analysis of nasal polyps (NP) with rich eosinophils (Eos) (n = 5) or poor Eos (n = 5), or without Eos (n = 5). CCL4 and EpCAM expression levels were measured. CCL4 (green), EpCAM (orange), MBP (red), and the nucleus (blue) are stained with hematoxylin and eosin (HE) (i). Images were obtained using an FV3000 confocal microscope (100 × objectives). The scale bars in the bottom-right corner indicate 100 μm. CCL4 intensity is expressed as a fold change relative to EpCAM (ii). The correlation of CCL4 expression with eosinophil count in NPs (iii) and Eos from the peripheral blood (iv). (**B**) *CCL4*, *CCL5*, *CCL11*, and *CCL26* mRNA levels were determined in NPs and uncinate process tissues (UTs) obtained from patients with ECRS. (**C**,**D**) CCL4 expression in BEAS-2B cells co-cultured with purified Eos (from peripheral blood) overnight. *CCL4*, *CCL5*, *CCL11*, and *CCL26* mRNA levels were determined (**C**). Images (CCL4, green; nucleus, blue) were obtained using an FV3000 confocal microscopes (200 × objectives) (**D**). Scale bars in the bottom-right corner indicate 10 μm. The values in B and C represent the mean ± SEM of four experiments. ^#^
*p* < 0.05, ^##^
*p* < 0.01 (vs. NPs with rich Eos in A, UTs from the same patients in B, and NPs without Eos in C).

**Figure 2 ijms-23-16149-f002:**
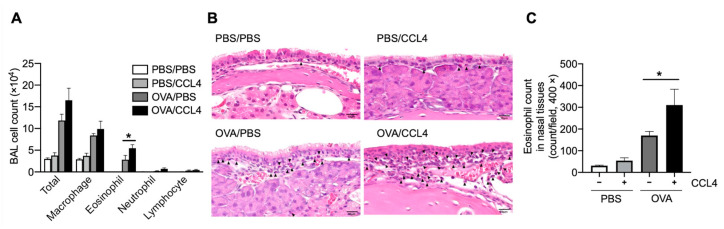
Effect of CCL4 on eosinophil accumulation in OVA-induce allergic model. Before each OVA challenge, recombinant murine CCL4 was intranasally administrated. (**A**) Total and differential cell counts in BALF. (**B**,**C**) Eosinophil infiltration in nasal tissues. HE staining (**B**) and the number of eosinophils infiltrated into the nasal tissues (**C**) are shown. Eosinophil infiltrates are indicated using black arrowheads. The scale bars in the bottom-right corner indicate 10 μm. The values represent the mean ± SEM of four (**A**) or three (**C**) experiments. * *p* < 0.05 (between the two groups). The experimental design is detailed in the Materials and Methods section (4.9. Sensitization and airway challenge).

**Figure 3 ijms-23-16149-f003:**
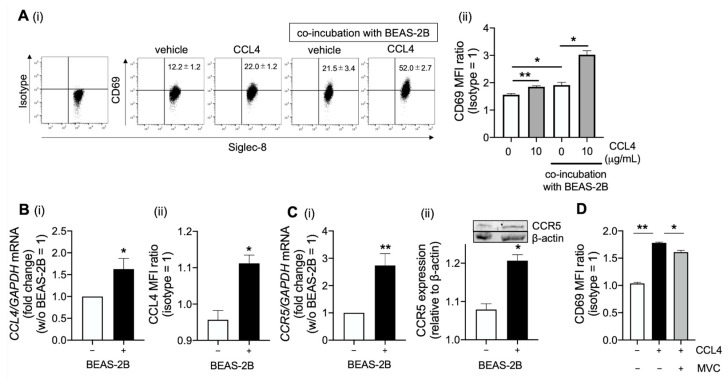
CCL4-mediated eosinophil activation. (**A**) Purified peripheral blood eosinophils coincubated with BEAS-2B cells were treated overnight with human recombinant CCL4. CD69 expression in eosinophils was evaluated [(i) percentage of CD69+ Siglec-8+ double-positive cells and (ii) CD69 median fluorescence intensity (MFI) ratio to isotype control]. (**B**) CCL4 mRNA (i) and protein (ii) expression in eosinophils coincubated with BEAS-2B cells. (**C**) CCR5 mRNA (i) and protein (ii) expression in eosinophils coincubated with BEAS-2B cells. (**D**) Eosinophils pretreated with maraviroc (MVC)—a CCR5 antagonist—were stimulated with CCL4. The values represent the mean ± SEM of four experiments. * *p* < 0.05, ** *p* < 0.01 (between the two groups).

**Figure 4 ijms-23-16149-f004:**
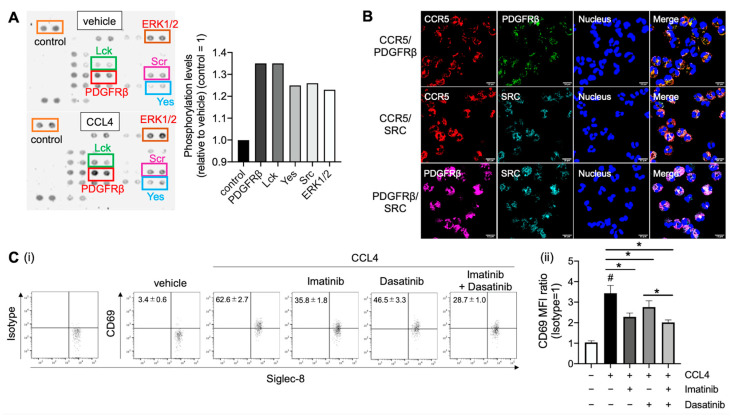
CCL4-mediated eosinophil activation signaling. (**A**) Phosphorylation levels of multiple kinases in eosinophils. Purified peripheral blood eosinophils were treated with CCL4 (10 μg/mL) for 5 min using a Human Phospho-Kinase Antibody Array Kit (R&D). Phosphorylation levels of platelet-derived growth factor receptor (PDGFR)β, Src kinase family (Lck, Src, and Yes), extracellular signal-regulated kinase (ERK), and control are presented as dots (left panel) or a relative value to vehicle (right panel). (**B**) Colocalization of CCR5, PDGFRβ, and Src in eosinophils. CCR5 (red), PDGFRβ (green or pink), Src (light blue), and the nucleus (blue) are shown. Images were obtained using an FV3000 confocal microscope (600 × objectives). The scale bars in the bottom-right corner indicate 10 μm. The results are representative of at least three experiments. (**C**) Purified peripheral blood eosinophils were pretreated with imatinib—a PDGFRβ inhibitor (I, 5 μM) —and/or dasatinib—a Src inhibitor (D, 10 nM) —for 30 min, followed by overnight incubation with CCL4 (10 μg/mL). CD69 expression in eosinophils was determined [(i) percentage of CD69+ Siglec-8+ double-positive cells and (ii) CD69 median fluorescence intensity (MFI) ratio to isotype control]. The values represent the mean ± SEM of four experiments. ^#^
*p* < 0.05 (vs. nontreatment control); * *p* < 0.05 (between the two groups).

**Figure 5 ijms-23-16149-f005:**
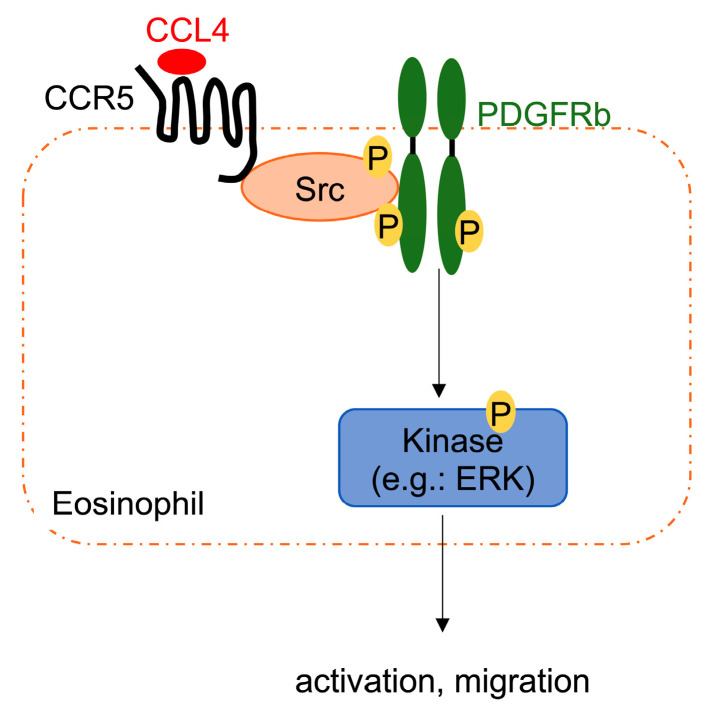
Mechanism of CCL4-mediated transactivation in the eosinophil signaling pathway. CCR5—a transmembrane protein coupled to G proteins that is stimulated with CCL4—induces PDGFRβ transactivation via tyrosine phosphorylation of Src, which leads to biological activities in eosinophils via signal transduction, such as kinase (e.g., ERK) phosphorylation.

## Data Availability

Not applicable.

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
