# Peer review of "CCL4 Regulates Eosinophil Activation in Eosinophilic Airway Inflammation"

_ijms, 2022, doi:10.3390/ijms232416149_

Round 1

Reviewer 1 Report

The manuscript by Chu et al. focuses on investigating the effect of the chemokine CCL4 in eosinophil activation in the context of eosinophilic airway inflammation.

The results show CCL4 is expressed in EpCAM-expressing human nasal polyps and that this expression positively correlates with eosinophils level in the patient's tissue. In a model of OVA-induced airway inflammation, the authors observed that co-administration of CCL4 increased eosinophil numbers in BALF. Further, the authors showed that treating human eosinophils with CCL4 and co-incubation with BEAS-2B cells further increased the expression of the activation marker CD69 and that using a CCR5 antagonist partially decreased CD69 expression on CCL4-treated eosinophils. Finally, the authors show that human eosinophils express PDGFRb and Src kinases, and treating eosinophils with CCL4 enhanced the phosphorylation of these kinases. Inhibition of PDGFRb or Src partly reduced CD69 expression in human eosinophils after treatment with CCL4. The manuscript shows exciting data regarding the involvement of CCL4 in the activation of eosinophils and patients with ECRS. However, it would significantly improve if the authors could address the following points:

Major comments: 

  1. Although referenced in the bibliography, the authors should include a statement of the clinical or laboratory criteria to define a "rich" versus a "poor" eosinophil infiltrate in nasal polyps.
  2. Figure 1 would benefit if they could show the merged images in panel 1A as they did in Fig. 4B.
  3. The experiments in Figure 3 show the synergic effect of co-culturing BEAS-2B cells + CCL4 on eosinophil activation. However, in figure 2D, the impact of the CCR5 antagonist on CD69 expression is shown on eosinophils without BEAS-2B cells. Although statistically significant, the CCR5 antagonist slightly reduced CD69 expression on eosinophils. Did the authors try this experiment in conjunction with BEAS-2B cells? What effect would they expect if MVC was used in the co-culturing experiments?
  4. As in the previous point, the experiments in Figure 4C were done only with eosinophils treated with CCL4. Did the authors attempt to inhibit PDGFRb and Src in the presence of BEAS-2B cells? What would be the outcome of this experiment?
  5. Can the authors explain why the expression levels of CD69+ cells after overnight CCL4 treatment range from 22% in Fig. 3A to 62%in Fig. 4C?
  6. Can the authors clarify their experimental design for the ex vivo assays? The figure legend says the data shown are the mean values of four experiments. Still, it is not clear whether this means cells from four different volunteers or experiments done four times with one biological replicate.
  7. In lines 220-221, the authors state that "PDGFRb inhibition sufficiently restored CCL4-mediated eosinophil activation, while Src inhibitor partially restored it"; the data shown in Figure 4C does not support these conclusions.
  8. The demographics of Table S2 show that most of the patients analyzed were males, whereas the mouse studies were done in females; is there evidence of a gender or sex bias in patients with ECRS?
  9. Do the soluble CD69 and CCL4 levels in serum and mucin in Figure S1A, B correlate with rich/poor eosinophil levels?
  10. Figure S1C shows a clear biphasic pattern between soluble CCL4 and CD69. Does this correlate with rich vs. poor eosinophil levels in ECRS patients?

Minor comments: 

  • Line 52: the authors state that there are eosinophils subpopulations, but they don't list them.
  • Figure 2: the figure legend is missing many details about the experimental design of the mouse model (time point, etc.)
  • Figure 2B: missing scale bar.
  • Line 258: the authors should state which markers they used to immunophenotype eosinophils and the other immune cells in the human and mouse studies.
  • Line: 206: the word "extremely" seems out of place and does not translate to a quantity or measurable unit.
  • Line 240: How many hours does "overnight" culture mean?
  • Line 255: details are missing about how the human samples were collected (fixed, sectioned, etc.) for immunofluorescence.
  • Line 260: can the authors briefly describe how they evaluated the ratio intensity in ImageJ?
  • Line 270: in the spirit of reproducibility, can the authors provide the clones of the flow cytometry antibodies used?
  • The details for how serum and mucin were obtained, processed, and measured for CCL4/CD69 are missing from the Methods section.

Author Response

We would like to thank for Reviewer’s favorable comments. We tried to answer the questions.

Major Comments:

  1. Although referenced in the bibliography, the authors should include a statement of the clinical or laboratory criteria to define a "rich" versus a "poor" eosinophil infiltrate in nasal polyps.

(Response)

In this study, we defined as “rich” if the number of eosinophils in submucosa of nasal polyps is higher than 100/high power field (HPF) and as “poor” if that is less than 100/HPF. We added this information in the Results section (2.1)

  1. Figure 1 would benefit if they could show the merged images in panel 1A as they did in Fig. 4B.

(Response)

As reviewer suggested, we replaced the pictures of NP with rich eosinophils with better pictures and added the merged images in panel 1A.

  1. The experiments in Figure 3 show the synergic effect of co-culturing BEAS-2B cells + CCL4 on eosinophil activation. However, in figure 2D, the impact of the CCR5 antagonist on CD69 expression is shown on eosinophils without BEAS-2B cells. Although statistically significant, the CCR5 antagonist slightly reduced CD69 expression on eosinophils. Did the authors try this experiment in conjunction with BEAS-2B cells? What effect would they expect if MVC was used in the co-culturing experiments?

(Response)

As we focused on simple signaling through CCR5 on eosinophils, we did not try this experiment. Because synergic effect may partially depend on enhancement of CCL4 release and CCR5 expression, MVC would partially inhibit CD69 expression also in conjunction with BEAS-2B cells.

  1. As in the previous point, the experiments in Figure 4C were done only with eosinophils treated with CCL4. Did the authors attempt to inhibit PDGFRb and Src in the presence of BEAS-2B cells? What would be the outcome of this experiment?

(Response)

In this study, we simply wanted to evaluate CCL4-mediated signaling pathway in eosinophils. Therefore, we did not attempt these experiments.

  1. Can the authors explain why the expression levels of CD69+ cells after overnight CCL4 treatment range from 22% in Fig. 3A to 62%in Fig. 4C?

(Response)

For the experiments in Fig. 3A, we used purified eosinophils. On the other hand, we used whole blood cells for the experiments in Fig. 4C because the procedure of eosinophil purification could change signaling status. Therefore, the interaction between other cells may affect CD69 expression on eosinophils.

  1. Can the authors clarify their experimental design for the ex vivo assays? The figure legend says the data shown are the mean values of four experiments. Still, it is not clear whether this means cells from four different volunteers or experiments done four times with one biological replicate. 

(Response)

 Basically, the cells were from different volunteers (or mixed with one or two biological replicates in some experiments). As we mentioned in Materials and Methods section, eosinophils were isolated from the peripheral blood of healthy volunteers with mild eosinophilia (approximately 4%–8% of total white blood cells) to uniform the character of eosinophils.

  1. In lines 220-221, the authors state that "PDGFRb inhibition sufficiently restored CCL4-mediated eosinophil activation, while Src inhibitor partially restored it"; the data shown in Figure 4C does not support these conclusions. 

(Response)

As reviewer suggested, we changed the phrase as follows;

“compared with Src inhibition, PDGFRβ inhibition significantly restored CCL4-mediated eosinophil activation to a greater extent”

  1. The demographics of Table S2 show that most of the patients analyzed were males, whereas the mouse studies were done in females; is there evidence of a gender or sex bias in patients with ECRS?

(Response)

 As previous studies revealed that eosinophilic inflammation can be induced in female mice to a greater extent than male mice in OVA-induced airway inflammation model, we used female mice in this study. Regarding incidence rate or severity of ECRS, there are no gender differences.

  1. Do the soluble CD69 and CCL4 levels in serum and mucin in Figure S1A, B correlate with rich/poor eosinophil levels?

(Response)

 The samples were obtained from different patients whose nasal polyps were not analyzed. Unfortunately, we do not have any data about these correlations.

  1. Figure S1C shows a clear biphasic pattern between soluble CCL4 and CD69. Does this correlate with rich vs. poor eosinophil levels in ECRS patients?

(Response)

 As the same with above response, the samples were obtained from different patients whose nasal polyps were not analyzed. Unfortunately, we do not have any data about these correlations.

Minor Comments:

  1. Line 52: the authors state that there are eosinophils subpopulations, but they don't list them.

(Response)

There are two subpopulations (resident eosinophils in steady-state and inducible eosinophils under pathological condition). We added this information as follows;

“eosinophils are classified into two subpopulations: resident and inducible eosinophils.”

  1. Figure 2: the figure legend is missing many details about the experimental design of the mouse model (time point, etc.).

(Response)

We described details of the experimental design in the Materials and Methods (4.9). We added this information in the legend of Figure 2 as follows;

“The experimental design is detailed in the Materials and Methods section (4.9. Sensitization and airway challenge).”

  1. Figure 2B: missing scale bar.

(Response)

We added scale bar in Figure 2B.

  1. Line 258: the authors should state which markers they used to immunophenotype eosinophils and the other immune cells in the human and mouse studies.

(Response)

EpCAM is used for a marker of epithelial cells and MBP is used for a maker of eosinophils. We added this information.

  1. Line: 206: the word "extremely" seems out of place and does not translate to a quantity or measurable unit.

(Response)

As reviewer suggested, we replaced it with “significantly”.

  1. Line 240: How many hours does "overnight" culture mean?

(Response)

 In this study. Overnight means 20-24 hours. We added this information.

  1. Line 255: details are missing about how the human samples were collected (fixed, sectioned, etc.) for immunofluorescence.

(Response)

 As reviewer suggested, we added some information in the Materials and Methods section (4.3) as follows;

“Nasal polyps were formalin-fixed and paraffin-embedded following endoscopic sinus surgery under general anesthesia.”

  1. Line 260: can the authors briefly describe how they evaluated the ratio intensity in ImageJ?

(Response)

 Briefly, all CCL4 or EpCAM signals were labeled, and the intensity was evaluated using ImageJ (a single signal was reversed to 8-bit). We briefly described it in the Materials and Methods section (4.3).

  1. Line 270: in the spirit of reproducibility, can the authors provide the clones of the flow cytometry antibodies used?

(Response)

 As reviewer suggested, we provided the clones information in the Material and Methods section.

  1. The details for how serum and mucin were obtained, processed, and measured for CCL4/CD69 are missing from the Methods section.

(Response)

 As reviewer suggested, we added some information in the Materials and Methods section as follows;

“4.10. CCL4 and soluble CD69 immunoassay

The CCL4 and souble CD69 levels in serum and mucin supernatants were measured using ELISA kits for MIP-1β/CCL4 (R&D Systems, Minneapolis, MN, USA) and soluble CD69 (Aviscera Bioscience, Santa Clara, CA, USA). Serum samples were collected from healthy volunteers and patients with refractory ECRS. Furthermore, mucin samples from patients with refractory ECRS were dissolved in PBS containing 0.1% dithiothreitol.”

Author Response

We would like to thank for Reviewer’s favorable comments.